# Generation of Sesame Mutant Population by Mutagenesis and Identification of High Oleate Mutants by GC Analysis

**DOI:** 10.3390/plants12061294

**Published:** 2023-03-13

**Authors:** Ming Li Wang, Brandon Tonnis, Xianran Li, John Bradly Morris

**Affiliations:** 1Plant Genetic Resources Conservation Unit, USDA-ARS, 1109 Experiment Street, Griffin, GA 30223, USA; 2Wheat Health, Genetics, and Quality Research, USDA-ARS, 291 Clark Hall, Pullman, WA 99164, USA

**Keywords:** *Sesame indicum*, EMS-mutagenesis, high oleate mutant, *FAD2* gene sequencing, breeding improvement, seed nutrition quality

## Abstract

Sesame is one of the important oilseed crops in the world. Natural genetic variation exists in the sesame germplasm collection. Mining and utilizing the genetic allele variation from the germplasm collection is an important approach for seed quality improvement. The sesame germplasm accession, PI 263470, which has a significantly higher level of oleic acid (54.0%) than the average (39.5%), was identified by screening the entire USDA germplasm collection. The seeds from this accession were planted in a greenhouse. Leaf tissues and seeds were harvested from individual plants. DNA sequencing of the coding region of the fatty acid desaturase gene (*FAD2*) confirmed that this accession contained a natural mutation of G425A which may correspond to the deduced amino acid substitution of R142H leading to the high level of oleic acid, but it was a mixed accession with three genotypes (G/G, G/A, and A/A at the position). The genotype with A/A was selected and self-crossed for three generations. The purified seeds were used for EMS-induced mutagenesis to further enhance the level of oleic acid. A total of 635 M_2_ plants were generated from mutagenesis. Some mutant plants had significant morphological changes including leafy flat stems and others. M_3_ seeds were used for fatty acid composition analysis by gas chromatography (GC). Several mutant lines were identified with high oleic acid (70%). Six M_3_ mutant lines plus one control line were advanced to M_7_ or M_8_ generations. Their high oleate traits from M_7_ or M_8_ seeds harvested from M_6_ or M_7_ plants were further confirmed. The level of oleic acid from one mutant line (M_7_ 915-2) was over 75%. The coding region of *FAD2* was sequenced from these six mutants, but no mutation was identified. Additional loci may contribute to the high level of oleic acid. The mutants identified in this study can be used as breeding materials for sesame improvement and as genetic materials for forward genetic studies.

## 1. Introduction

Sesame is an important oilseed crop with a long history of cultivation (over 3000 years). Seeds, leaves, and oil from sesame have been utilized and consumed by humans as a vegetable and food ingredient for about 6000 years [1]. Sesame seeds contain 48–55% oil, 20–28% protein, 14–16% sugars, 6–8% fibers, and other nutritional and/or bioactive compounds with beneficial effects to human health such as vital minerals, vitamins, phytosterols, tocopherols and lignans [2]. The beneficial effects to human health from these bioactive compounds include the prevention of degenerative diseases such as cancer, cardiovascular diseases, atherosclerosis, and the process of aging. In the traditional Chinese and Indian systems of medicine, sesame seed and oil also played an important role in treatments such as rubbing sesame oil on the skin to sooth minor burns, aid in healing chronic skin disease, and consumption of roasted sesame seeds by new moms for producing breast milk. Significant variation for oil content, protein content, lignan, tocopherol, and fatty acid concentrations was found among eight sesame genotypes. Oil, protein, sesamin and sesamolin, tocopherol, oleic and linoleic, and the minor unsaturated fatty acids were found ranging from 29.43 to 54.69%, 13.92 to 21.76%, 0.55 to 8.98 mg/g, 0 to 239.58 µg/g, 26.6 to 54.85%, and 0.13 to 0.89%, respectively [3]. Since sesame is a highly nutritional and valuable crop, some basic genetic and genomic research work has been conducted on sesame for exploring its utilization potential and developing new cultivars.

Cultivated sesame (*Sesamum indicum* L.) is a diploid species (2*n* = 2*x* = 26) with a relatively small nuclear genome size (*n* = 354 Mb) [4,5]. Its genome has been sequenced using Illumina technology [6]. Some candidate genes involved in oil biosynthetic pathways have been identified by the analysis of comparative genomics and transcriptomics [7,8], and an integrated database (i.e., genetic information combined with comprehensive phenotypic information) for the functional genomics of sesame (SesameFG) has also been made publicly available [9]. Recently, gene-editing using the CRISPR/Cas9 (clustered regularly interspaced short palindromic repeats-associated protein 9) system with hair root transformation was also successfully applied for the targeted mutagenesis of genes in sesame for sesamin and sesamolin biosynthesis [10].

Improving seed nutritional quality (for example, elevation of oleic/linoleic acid ratio, O/L) is one of the main objectives for oilseed crop breeding programs. Seeds with the high O/L ratio (i.e., high oleic acid content and low linoleic acid content) can not only increase the seed shelf-life for preservation but also provide high quality sesame products beneficial to human health including reducing blood pressure and risk of cardiovascular diseases. High oleate (about 80%) cultivars/traits have been developed and identified in several major oilseed crops such as peanuts, soybeans, canola, and safflower [11,12,13,14] through breeding programs. However, little work has been reported in identification and development of high oleate cultivars in sesame except for some preliminary sesame mutagenesis work on other traits [15,16]. Screening the germplasm collection and EMS-mutagenesis are two efficient approaches, respectively, for identification of naturally occurring high oleate mutants and creation of newly induced high oleate mutants. Our goal was to develop high oleate germplasm lines in sesame by using these two approaches synergistically (i.e., first identification of a natural high oleate mutant and then mutagenesis of this natural high oleate mutant for further enhancing the oleic acid level to about 80%). Therefore, the objectives of this study are to (1) screen the USDA sesame germplasm collection to identify natural high oleate accessions, (2) conduct mutagenesis of the natural high oleate mutant to further enhance the oleic acid level, and (3) characterize the high oleate mutants to generate useful genetic materials for breeders and geneticists to use for development of new sesame cultivars and identification of corresponding genes for the high oleate trait.

## 2. Results and Discussion

### 2.1. Identification of Naturally High Oleate Mutants from the Germplasm Collection

#### 2.1.1. Identification of Naturally High Oleate Mutants by GC Analysis

One accession (PI 263470) containing a significantly higher level of oleic acid (54.0%) than the average (39.4%) (unpublished data, Wang et al.) was identified from the USDA sesame germplasm collection by gas chromatography (GC) analysis (Figure 1A). From the germplasm passport record, this accession originated in the former Soviet Union and was donated to the collection in 1960 by Toyama University, Japan. The high oleate trait was followed up by further GC analysis and DNA sequence analysis using fresh seeds and leaf tissue collected from the plants regrown in the greenhouse.

#### 2.1.2. Confirmation of Naturally High Oleate Mutants by DNA Sequence Analysis

The *FAD2* gene encodes a fatty acid desaturase (FAD) that is responsible for the desaturation ratio of oleic acid to linoleic acid (O/L) in oilseed crops. To check and compare the coding sequence of the *FAD2* gene among individual plants of PI 263470 plus PI 224663 (as a control for the average level of oleic acid), leaf tissues were collected from individual plants. The results from DNA sequence analysis of multiple plants indicated that position 425 in the coding sequence had A/A, G/A, and G/G genotypes (Figure 2 and Appendix A). Thus, DNA sequences from individual plants confirmed that PI 263470 was a mixed accession.

The results from a comparison of *FAD2* gene sequences among eight sesame accessions and partially deduced amino acid sequences of *FAD2* among sesame, soybean, and peanut are shown in Appendix A. Three histidine (H)-boxes (critical sites for fatty acid desaturase displaying its function) were identified, but only the flanking sequences around H-Box2 is shown in Appendix A. The DNA mutation (G425A) in PI 263470 (Appendix A) occurred in H-Box2, which led to an amino acid substitution of R142H (arginine substituted to histidine) within the fatty acid desaturase. This amino acid substitution likely explains the higher oleate content in seeds of PI 263470 relative to other sesame accessions. Therefore, this functional mutation can presumably reduce fatty acid desaturase activity and result in an increased level of oleic acid and the decreased level of linoleic acid in sesame seeds.

#### 2.1.3. Reconfirmation of Naturally High Oleate Mutants by GC Analysis

Additional seeds from eight individually sequenced plants were collected and used for GC analysis. The results from GC analysis are shown in Figure 2 and Appendix A. The heterozygous genotype of plant 3-6 (A/G) and homozygous genotype of plant 3-11 (A/A) from PI 263470 had 47.2 and 56.6% oleic acid, respectively. They had a higher level of oleic acid than the homozygous genotype of plant 2 (G/G) from PI 224663 with 39.22% oleic acid. All other plants (3-1, 3-3, 3-10, 3-22, and 3-23) with the A/A genotype from PI 263470 also produced seeds with >52% oleic acid. The results from both sequence analysis and GC analysis suggest that the natural mutation of G425A in the coding region of the *FAD2* gene may be responsible for the enhanced level of oleic acid from 39.2 to 56.6%.

### 2.2. EMS Mutagenesis and Generation of Mutant Population

#### 2.2.1. EMS Mutagenesis

In other oilseed species, the oleate level can reach to about 80%. The natural mutant sesame accession we identified was only about 54–56%. This accession would be a good starting genetic material for mutagenesis to further enhance the level of oleate.

The germination rate and seed setting can be greatly affected by EMS treatment dosage. To determine the EMS concentration, four dosages were used (phosphate buffer only, 1.0%, 1.5%, and 2.0%). We found after EMS treatment, the seed germination rate (95.31%) with 1.0% EMS treatment was very close to the control (no EMS) germination rate (95.75%). After further reducing EMS concentration, we concluded that 0.8% EMS concentration may be the most suitable treatment for sesame seed mutagenesis for our lab protocol.

#### 2.2.2. Generation of Mutant Populations

In 2014, 1800 sesame seeds were treated with 0.8% EMS. The procedure of mutagenesis from M_1_ seeds to M_3_ seeds is shown in Figure 3. After planting in the greenhouse, 1553 M_1_ seeds germinated for a rate of 86.3%. Of these, some seedlings died, and some mature plants did not produce seeds. In the end, we only harvested seeds from 1412 M_1_ plants for a seed setting rate of 90.9%. In 2015, one or two seeds from each M_2_ line (depending on the seeds available for each line) were planted to clay pots. After planting, 1030 M_2_ seeds were germinated. The average germination rate of M_2_ seeds was 72.95%. After germination, some seedlings either died or the mature plants did not produce seeds despite flowering. Only 329 M_2_ plants produced M_3_ seeds. The M_2_ plants seed setting rate (31.94%) was probably reduced because more recessive lethal alleles were revealed or exposed in M_2_ from M_1_ selfing. M_2_ plant morphology was observed in the greenhouse. M_3_ seeds were harvested and used for chemical analysis, genetic analysis, and planted for advancement to the next generation.

### 2.3. Characterization and Evaluation of Mutants

#### 2.3.1. Morphological Mutants

Although 1030 M_2_ seeds germinated in the greenhouse, some M_2_ seedlings showed albinism and some M_2_ seedlings did not develop into mature plants. In total, 405 M_2_ seedlings were dead before developing into mature plants. Among 625 M_2_ plants, some did not flower, while others flowered but did not produce seeds. Among M_2_ plants, clear variation from two agronomic traits (plant height and maturity) was observed. After GC analysis, seeds from some M_3_ lines were planted in one-gallon pots for morphological observation (Figure 4). Compared with the control (PI 263470), the M_3_ 200-1-11 plant had flat stems with darker leaves and M_3_ 447-1-1 was leafy (more leaves clustering at the top of plant). Morphological mutations with leafy and clustered capsules were also observed in a previous study [15]. Since identification of high oleate mutants was our focus in this study, only a few M_3_ mutant families were planted for further morphological observations in the early generations.

#### 2.3.2. DNA Sequence Analysis of High Oleate Mutants

Using GC analysis of M_3_ seeds from the M_2_ plants, we identified five high oleate mutants (M_3_ 965-2-36, M_3_ 965-2-33, M_3_ 965-2-11, M_3_ 915-1-35, and M_3_ 200-1-11). Their oleic acid level was approximately 70%. Leaf tissue was collected from M_3_ plants, and DNA was extracted from these leaves. These five high oleate mutants were sequenced in the coding region of the *FAD2* gene. Surprisingly, there were no mutations identified in the coding region when compared with the control (PI 263470 with A/A). We also attempted to sequence the promoter region of this gene but encountered repetitive sequence issues and could not get high quality sequence reads to assemble the complete sequence of the promoter region. DNA sequencing from more genome regions is required.

Two possible explanations may account for the increase in oleic acid content in mutants. First, the causal mutations are in the promoter region, but we have not identified any yet. Future experiments comparing the gene expression level of *FAD2* between the mutant and control may help to determine whether the regulatory region contributes to the enhanced level of oleic acid. Alternatively, other genes such as *ROD1* (reduced oleate desaturation 1) gene identified in *Arabidopsis* [17], but which has not been identified in sesame, may partially control oleic acid levels. These genes need to be identified and sequenced for comparison with the control line.

#### 2.3.3. High Generation of Oleate Mutant Lines

At the M_3_ generation, all identified lines were still segregating for fatty acid level and other plant morphological traits. We advanced these mutant lines in the greenhouse while monitoring the level of oleic acid as well as conducting some minor selection on plant morphology and maturity. In 2022, six high oleate mutant lines plus the control were planted (28 April 2022) and grown in the greenhouse. Their planting, flowering, and harvest dates are listed in Appendix A. Compared with the control (PI263470 A/A), five mutants flowered at similar times (around 20 May), but the mutant M_7_ 965_2_33_7_6_10 had small plant stature at the early stage (about one month old) and flowered on 06/28, which was over a month later than the control (Figure 5). Two mutants (M_7_ 915_1_35_4_3_2 and M_7_ 915_1_35_5_5_3) were harvested three and five days earlier than the control, respectively (07/18). Four mutants (M_7_ 965_2_36_5_3_36, M_7_ 965_2_11_6_2_29, M_8_ 200_1_11_15_3_5_21, and M_7_ 965_2_33_7_6_10) were harvested six days, eight days, ten days, and one month later than the control, respectively.

The seed fatty acid composition results from each mutant line plus the control are summarized in Appendix A and shown in Figure 6. There were 12 fatty acids identified. Among them, four major fatty acids (>1%) were palmitic (C16:0), stearic (C18:0), oleic (C18:1), and linoleic (C18:2) [18,19]. The rest were minor fatty acids (<1%). For each fatty acid, the average value from eight individual plants is in the cell, and the range is in the bracket for each genotype in Appendix A. For example, the average value and range of stearic acid (C18:0) for the control (PI 263470) is 5.28% and 5.09–5.44%, respectively.

For statistical analysis, each fatty acid value from all mutants was compared with the control value, and the results are presented in Figure 7. Six different color dots represent six mutant lines. The size of different dots represents the different percentage of fatty acids. Larger dots correspond to higher fatty acid percentages. The lower and upper dotted lines represent the threshold value for *p* = 0.05 and *p* = 0.01, respectively. The palmitic acid (C16:0) levels (6.80%, 7.48%, 8.32%, and 9.24%) from four mutants (M_8_ 200-1, M_7_ 915-2, M_7_ 915-3, and M_7_ 965-10) were significantly lower than the control (9.54%) at *p* = 0.01, but the palmitic acid level (9.40% and 9.41%) from two mutants (M_7_ 965-29 and M_7_ 965-36) was not significantly different from the control at *p* = 0.05. The stearic acid (C18:0) levels (7.20%, 7.08%, and 7.81%) from three mutants (M_8_ 200-1, M_7_ 915-2, and M_7_ 915-3) were significantly higher than the control (5.28%) at *p* = 0.01. The stearic acid levels (4.65% and 5.05%) from M_7_ 965-10 and M_7_ 965-29 were significantly lower than the control (5.28%) at *p* = 0.01 and *p* = 0.05, respectively. The stearic acid level (5.50%) from mutant M_7_ 965-36 was not significantly different from the control. The oleic acid (C18:1) and linoleic acid (C18:2) levels from all mutants were significantly higher and lower than the control at *p* = 0.01, respectively (Appendix A and Figure 7). For the four major fatty acids, in general, stearic acid significantly increased and palmitic acid significantly decreased after mutagenesis. Additionally, oleic acid significantly increased and linoleic acid significantly decreased. Compared with the control, this large negative change between oleic acid and linoleic acid can be clearly observed from Figure 8. We also observed that the variation among eight individual plants within the same mutant was not significant.

#### 2.3.4. Correlation among Fatty Acids

The correlations among 12 fatty acids are shown in Figure 9. The *r* value can reflect how the traits change relative to one another. Most *r* values from the correlation of palmitoleic (C16:1), margaric (C17:0), and heptadecenoic (C17:1) with other fatty acids were lower than 0.5, whereas most *r* values among the other nine fatty acids were larger than 0.5. Therefore, we focused on discussing the correlations among these nine fatty acids from the top to bottom of Figure 9. Palmitic acid (C16:0) had significantly negative correlations (*r* = −0.77, −0.58, −0.48, −0.75, −0.90, −0.75, and −0.59) with stearic (C18:0), oleic (C18:1), linolenic (C18:3), arachidic (C20:0), eicosenoic (C20:1), behenic (C22:0), and lignoceric (C24:0) except for a significantly positive correlation (*r* = 0.58) with linoleic (C18:2) acid. Stearic acid had significantly positive correlations (*r* =0.52, 0.72, 0.96, 0.71, 0.85, and 0.76) with oleic, linolenic, arachidic, eicosenoic, behenic, and lignoceric except for a significantly negative correlation (*r* = −0.61) with linoleic. Oleic acid had significantly positive correlations (*r* = 0.52, 0.63, 0.51, 0.63, and 0.54) with linolenic, arachidic, eicosenoic, behenic, and lignoceric except for significantly and negatively correlated (*r* = −0.99) with linoleic acid. Linoleic acid had significantly negative correlations (*r* = −0.6, −0.72, −0.51, −0.69, and −0.61) with linolenic, arachidic, eicosenoic, behenic, and lignoceric. Arachidic acid had significantly positive correlations (*r* = 0.68, 0.93, and 0.88) with eicosenoic, behenic, and lignoceric. Eicosenoic acid had significantly positive correlations (*r* = 0.75 and 0.48) with behenic and lignoceric acids. Behenic acid also had a significantly positive correlation (*r* = 0.81) with lignoceric acid. The correlations among fatty acids identified above will be useful for sesame breeders to improve seed nutritional qualities in breeding programs. Furthermore, the *FAD2* genes showed a high effect for oleic acid and linoleic acid (or O/L ratio) in peanuts [20]. The trait for O/L ratio in sesame may also be highly heritable. Therefore, the high oleate trait can be easily introduced from the mutant lines into elite lines by backcrossing in sesame.

## 3. Materials and Methods

### 3.1. Plant Materials and Plant Growth under Greenhouse Conditions

Sesame seeds were planted in the greenhouse under controlled conditions (26.7 °C/8 h for day with natural light and 32.2 °C/16 h with natural dark). Due to the space limitation, we adjusted water content and fertilizer amount from planting to harvesting for controlling the plant height and size. The purpose was to harvest enough seeds from each plant for one generation. To prevent plants from lodging, each plant was supported by a bamboo pole tied with a piece of white plastic tape. The seeds were harvested from each capsule and put into small paper bags which were transferred to the seed storage room for drying (21 °C, 25% RH for two weeks). The dried seeds were used for next generation planting or fatty acid analysis.

### 3.2. EMS Mutagenesis

The accession (PI 263470) with high oleate (54%) was used for mutagenesis. On 3 March 2014, about 600 purified seeds harvested from the plant number 6-3-6 (with genotype A/A) were put into a 50 mL tube containing 40 mL of 100 mM phosphate buffer (pH 7.5) and were incubated at 4 °C for 24 h to allow seed imbibition. In total, we treated the seeds in four 50 mL tubes (4 tubes × 600 seeds = 2400 seeds). After imbibition, the buffer was removed. Freshly made 40 mL of 0.8% EMS in 100 mM phosphate buffer was added to the tube. The tube was covered with aluminum foil for preventing light followed by agitation on a shaker at 80 rpm at room temperature for 8 h. After EMS treatment, the EMS solution was removed into a waste container. The treated seeds were washed with tap water (with 3–4 changes during 30 min).

The treated (M_1_) seeds were planted into a tray filled with soil (Metro-Mix360, Product of Canada, Sun Gro Horticulture, Vassar, MB, Canada) and placed in the greenhouse. Each tray had 32 small pots, and two seeds were planted into each pot. After planting, the tray was covered with a transparent plastic cover to maintain moisture levels. After seed germination, the transparent plastic cover was removed immediately. Once the seedlings were well established (with two true leaves), they were thinned to one seedling per pot. This thinning process prevented us from calculating the mutation rate because we preferred to remain only one healthy plant per pot. When the plants grew taller, a thin bamboo pole was inserted into each pot and tied to the plant using a piece of white plastic tape to prevent plants from lodging (see M_1_ young seedlings and plants in Figure 3). On 21 April 2014, most plants flowered. To prevent plants from snapping, we watered the plants from both sides only. On 30 May 2014, some capsules turned brown and matured. At this time, we started harvesting mature capsules. We finished harvesting all the mature capsules by 7 July 2014. After drying, we found some capsules containing mature seeds and some capsules did not contain good seeds (immature or seeds with poor quality). After seed cleaning, 1412 lines of M_2_ seeds were harvested from the M_1_ plants.

On 11 August 2014, all M_2_ seeds plus the control (10 seeds from PI 263470 that were not treated with EMS) were planted into small clay pots (1–2 seeds planted in each pot) in the greenhouse. The M_2_ plant morphology was observed and recorded after 48 days of growth. Not all plants produced M_3_ seeds. M_3_ seeds were harvested from 329 M_2_ plants, and they were used for fatty acid composition analysis by GC.

Fatty acid analysis of M_3_ seeds showed five mutant lines with high oleate mutants (among them, one mutant line separated into two mutant lines due to significant segregation), and they were planted in the greenhouse and advanced to later generations. On 18 April 2022, six mutant lines (plus one control) were planted in the greenhouse. Plant morphology was observed and recorded. Seeds were harvested from eight plants for each mutant line and these seeds from individual plant were used for GC analysis.

### 3.3. Fatty Acid Composition Analysis with GC

Since M_3_ seeds were still segregating, individual M_3_ seeds were selected and analyzed for fatty acid composition using gas chromatography (GC). Two seeds were measured for two replicates. The average from two measurements represented the fatty acid composition value of each M_3_ line. In total, 635 M_3_ seeds were analyzed. A single seed was put into a 1.5 mL glass injection vial. A small glass rod was used for manually crushing the seed into a fine powder within the vial. Fatty acid methyl esters (FAMEs) were prepared from sesame seed powder by alkaline transmethylation. Fatty acid composition was determined on an Agilent 7890A gas chromatograph (Santa Clara, CA, USA) equipped with a flame ionization detector (FID) following a previously described lab method [21].

### 3.4. DNA Sequence and Deduced Amino Acid Sequence Analysis

Fresh leaf tissue (75–100 mg) was collected from individual sesame plants. DNA was extracted from these fresh leaf tissues using an Omega Bio-Tek E.Z.N.A. (Plant DNA kit, Norcross, GA, USA). DNA quality and quantity were determined on a Nanodrop 2000C spectrophotometer (Themo Scientific, Wilmington, DE, USA). DNA quality of samples was also checked by using 1% agarose gel electrophoresis. DNA concentration for all samples was diluted to 10 ng/µL. Diluted DNA was used as the template for amplification of the coding region of the *FAD2* gene. The primers (5′ TGGGAGGTTTTGATTCAGACA 3′ and 5′ CACTAACAAAGGACGCATCTTA 3′) from the flanking coding region of sesame *FAD2* were designed (modified from Chen et al., 2014) and synthesized by Operon (Qiagen, Germantown, MD, USA, http://www.operon.com, accessed on 18 November 2022). The PCR mix contained 2 mM of MgCl_2_, 0.5 mM of dNTPs, 0.25 µM each of primer, and 15 ng of DNA template for a total volume of 20 µL. The PCR cycling conditions consisted of 1 cycle of 94 °C for 5 min, 35 cycles of 94 °C for 30 s, 72 °C for 100 s, and a final extension of 72 °C for 7 min. Amplicons were sequenced using PCR primers and two internal primers (5′ GAGGCCAACCGAGAGTAAGA 3′ and 5′ CTGCTCGTGCCCTATTTCTC 3′) to cover the coding region. After trimming each sequence, consensus sequences were assembled using Sequencher, ver. 5.3. SNPs were identified from the natural mutant but not from EMS-induced mutants. Substitutions were identified in polypeptides by comparison of the deduced amino acid sequences.

### 3.5. High Generation of Mutant Lines

Five M_3_ mutant lines (M200-1-11, M915-1-35, M965-2-11, M965-2-33, and M965-2-36 plus the control were purified PI 263470 with A/A), further selected, and followed up to the M_7_ or M_8_ generations in 2022. Their M_6_ or M_7_ plant morphology was observed in the green house, and the oleate level from M_7_ or M_8_ seeds was measured by GC using 7–10 seeds in a bulk analysis.

### 3.6. Data Analysis

For the six mutant lines and control, the mean value from two replications (when available) of each fatty acid composition was analyzed and visualized with R v4.1.1. For each mutant, varTest in R was used to test whether the mutant had significantly different variance than the control. Then a student’s *t*-test was conducted to test whether the composition was significantly different between the mutant line and the control. Pairwise correlations among fatty acid composition were plotted by the pair R function.

## 4. Conclusions

We identified a natural high oleate accession (PI 263470) that harbors a nonsynonymous mutation (R142H) in the *FAD2* gene by screening the entire USDA sesame germplasm collection. This accession has 54% oleic acid compared with the average of 39.4% of the entire collection. Functional SNP markers were developed from this natural mutant for the marker-assisted selection (MAS). Because this high level of oleic acid in sesame is still lower than 80% achieved by high oleic varieties in other oilseed crops, we employed EMS-induced mutagenesis of this natural accession to further increase the oleic acid level. Several mutants that had oleic acid over 70% were isolated. However, no additional mutation was identified in the coding region of the *FAD2* gene. More genome regions (including regulatory regions and other possible copies of *FAD2* genes) should be sequenced. Crosses among these mutants can be made to combine different mutant loci if they are not linked to further enhance the oleic acid level. After reaching 80%, the new high oleate genetic lines can be crossed with the elite breeding lines to develop high oleate sesame cultivars.

## Figures and Tables

**Figure 1 plants-12-01294-f001:**
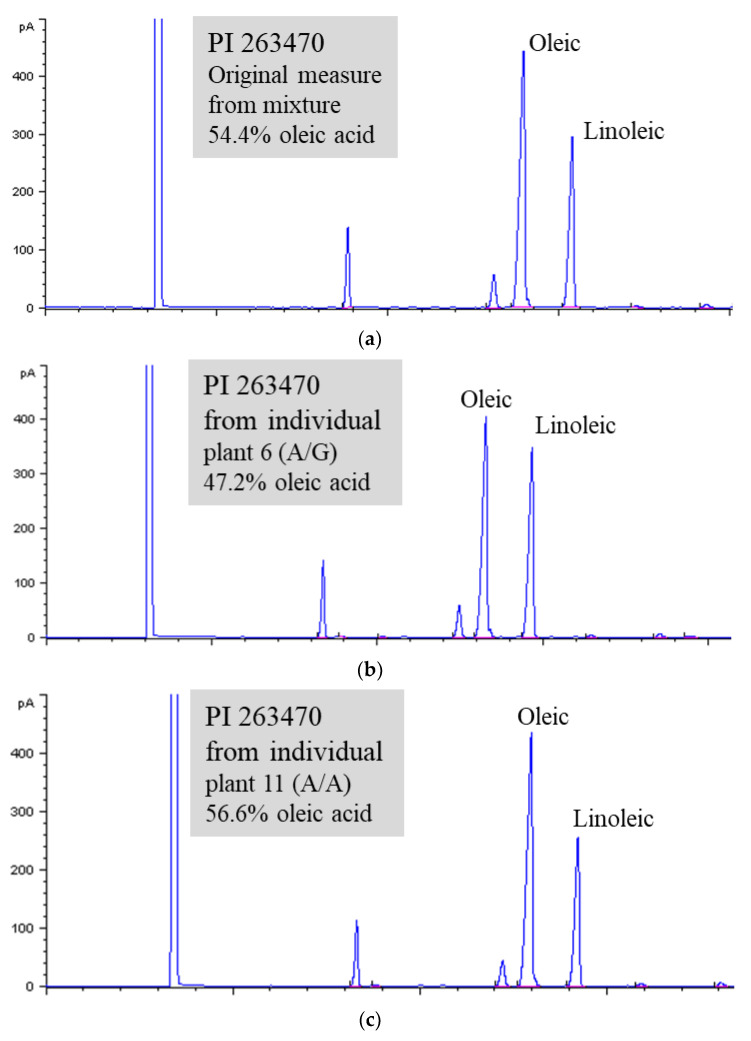
The levels of oleic acid and linoleic acid shown on chromatograms from three genotypes. (**A**). Mixed genotypes (A/A and A/G). (**B**). Genotype A/G. (**C**). Genotype A/A.

**Figure 2 plants-12-01294-f002:**
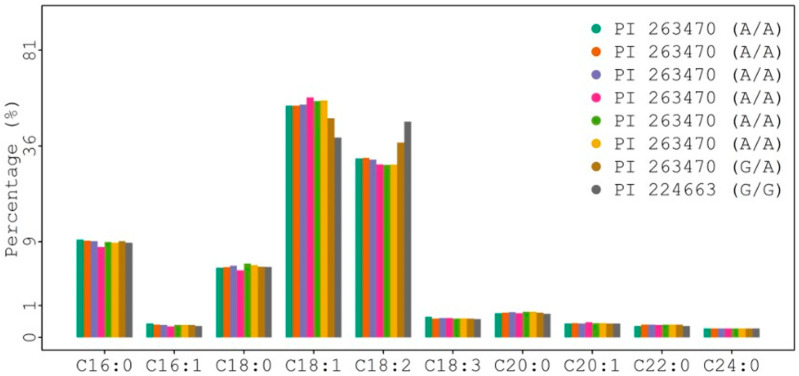
Comparison of fatty acid profiles among eight plants with *FAD2* genotypes.

**Figure 3 plants-12-01294-f003:**
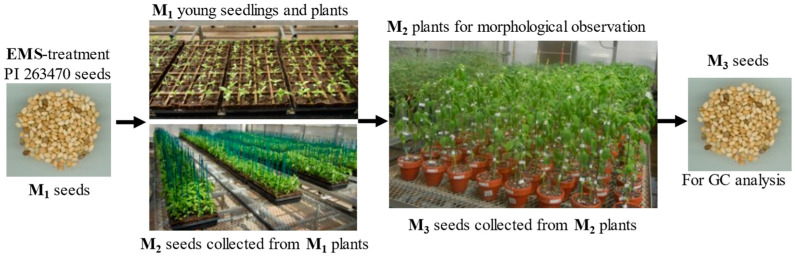
The procedure of mutagenesis from M_1_ seeds to M_3_ seeds.

**Figure 4 plants-12-01294-f004:**
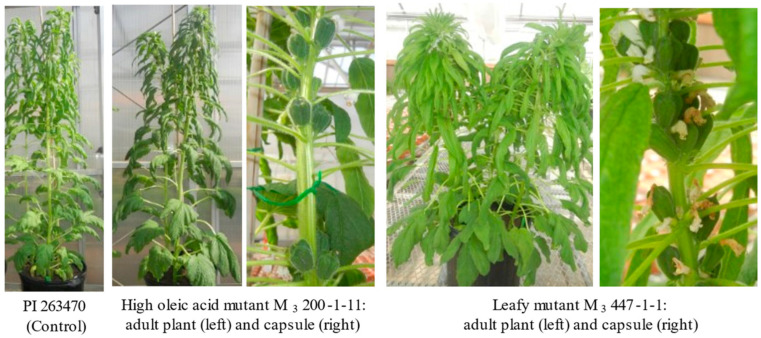
Comparison of plant morphology between mutant line and control.

**Figure 5 plants-12-01294-f005:**
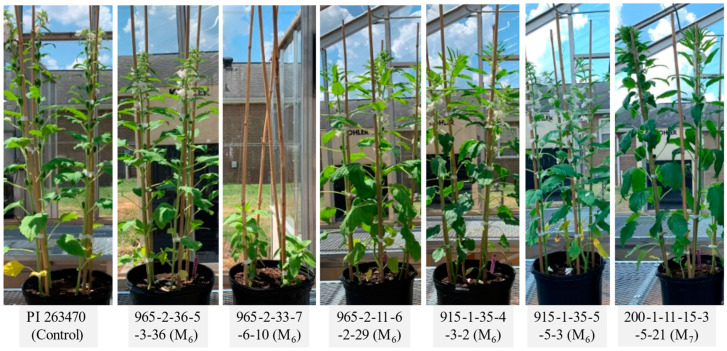
Comparison of plant morphology between mutant line and control at a high generations.

**Figure 6 plants-12-01294-f006:**
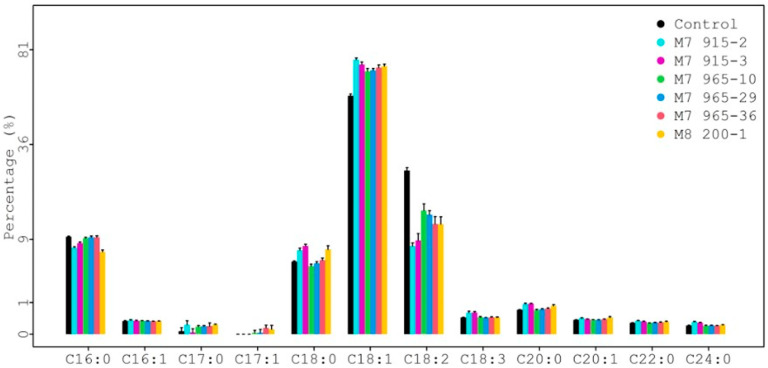
Comparison of fatty acid profiles among mutants at higher generations.

**Figure 7 plants-12-01294-f007:**
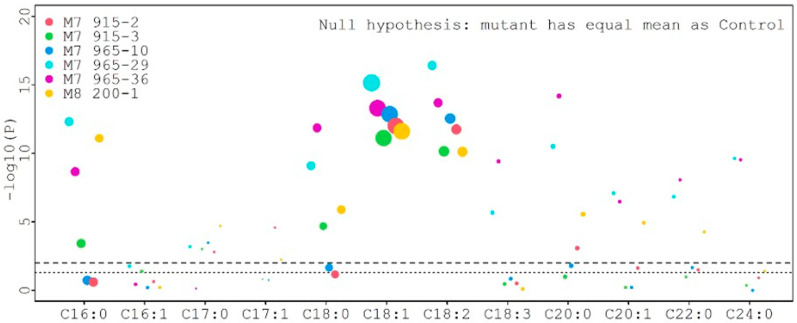
Comparison of each fatty acid value from all mutants with the control.

**Figure 8 plants-12-01294-f008:**
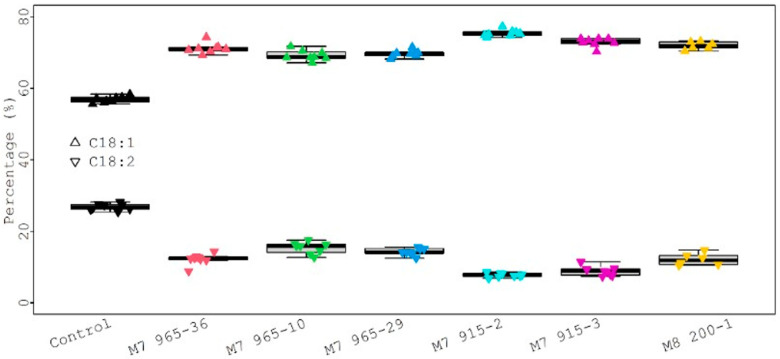
Comparison of oleic and linoleic between control and mutants.

**Figure 9 plants-12-01294-f009:**
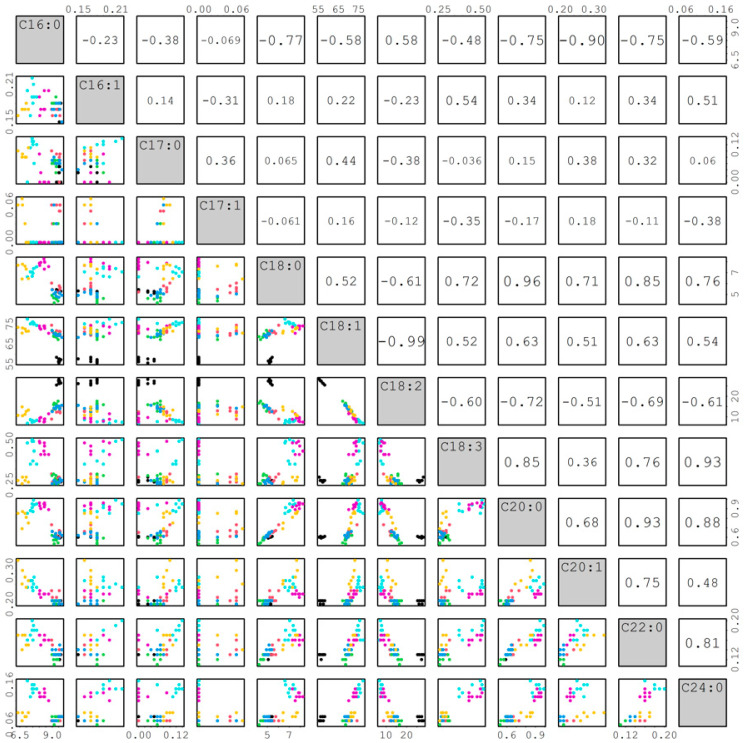
Correlations among different fatty acids in sesame.

## Data Availability

The data and plant materials that support the findings of this study are available from the corresponding author upon request.

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
