# Peer review of "Generation of Sesame Mutant Population by Mutagenesis and Identification of High Oleate Mutants by GC Analysis"

_plants, 2023, doi:10.3390/plants12061294_

Round 1
Reviewer 1 Report
For breeders to use the mutant, the heritability of the trait of Fat2 and the new non-identified locus need to be determined. Also, the other agronomic traits need to be evaluated for crossing with elite lines for developing new germplasm and cultivars.
Author Response
Response to Reviewer 1
12/28/2022
First, I would like to thank the reviewer for the comments and suggestions for enhancing the manuscript quality. For your convenience, all my changes would be in blue color text.
( ) I would not like to sign my review report
(x) I would like to sign my review report
English language and style
( ) English very difficult to understand/incomprehensible
( ) Extensive editing of English language and style required
( ) Moderate English changes required
(x) English language and style are fine/minor spell check required
( ) I don't feel qualified to judge about the English language and style
|
Yes |
Can be improved |
Must be improved |
Not applicable |
|
|
Does the introduction provide sufficient background and include all relevant references? |
(x) |
( ) |
( ) |
( ) |
|
Are all the cited references relevant to the research? |
(x) |
( ) |
( ) |
( ) |
|
Is the research design appropriate? |
(x) |
( ) |
( ) |
( ) |
|
Are the methods adequately described? |
(x) |
( ) |
( ) |
( ) |
|
Are the results clearly presented? |
(x) |
( ) |
( ) |
( ) |
|
Are the conclusions supported by the results? |
(x) |
( ) |
( ) |
( ) |
Comments and Suggestions for Authors
For breeders to use the mutant, the heritability of the trait of Fat2 and the new non-identified locus need to be determined. Also, the other agronomic traits need to be evaluated for crossing with elite lines for developing new germplasm and cultivars.
Response: Good point. One new reference (Pandey et al., 2014) was added as [20]. This reference showed FAD2 genes had high effect on oleic acid and linoleic acid (or O/L ratio) in peanuts. The trait for O/L ratio in sesame is a recessive trait that is highly heritable. Several sentences were added after line 261. This trait can be selected easily with the molecular marker designed according to the DNA polymorphism.
Submission Date
19 November 2022
Date of this review
12 Dec 2022 06:02:37

Reviewer 2 Report
The present manuscript proposes to generate mutant population by EMS and to identify high oleate mutants by GC analysis. This study is based on the use of a specific accession, that has already a “high oleate content” due to possible mutations in the FAD2 gene. Then, EMS mutagenesis is performed, followed by screening and finally the lipid composition of some mutant plants is analysed. While this work can have a significant interest for plant breeders, most of the conclusions draw by the study are not supported by the data. Indeed, “high oleate content” suggest that there is more oleate in terms of quantity. However, all the GC analysis showed % fatty acid profile, i.e. we do not have any information about the lipid quantity. Thus, differences in the fatty acid profile do not necessarily imply final differences on the fatty acid content. In fact, lipid analysis for fatty acid is always shown in terms of % and of quantity to disentangle this issue. Hence, the “high oleate content” promised in the study is questionable. The authors must show the quantitative results for all the mutant lines and discuss about it. Besides this, the authors proposed a link between a single point mutation and the activity of FAD2 with only one plant as A/G and one plant as G/G. I am not so convinced by these results and I think the authors should perform a backcross of an A/A line with a G/G line to obtain multiples lines with A/G pattern to confirm that the fatty acid profile corresponds to a mixture of A/A and G/G fatty acid profiles. This will also help the authors to obtain more biological replicates. Indeed, the statistical significance of the results can also be questioned: there are no biological triplicates for all the GC analysis, and some student’s t-test are performed with only two values per mean…Finally, the table 1 and 2 should be modified as bargraphs to be able to rapidly compare differences of fatty acid profiles. Regarding the absence of mutation in the FAD2 gene in the newly established mutant lines, I do not see any surprise. As the authors performed random mutations, many mutations in the genome of sesame can lead to a difference of the fatty acid profile without implying FAD2 gene. The authors also argued that some mutations in the promoter region of the FAD2 gene could occur in the mutant line and perhaps explained the phenotype for the fatty acid profile. Thus, the authors suggest that there is a differential expression of FAD2 gene in the mutant plants along seed filling. So can you please show us qPCR experiments for FAD2 gene in these mutants during seed filling ? Regarding the metadata about the phenotypical characterization of the mutants, I found it very short. I think it is necessary to have access to important parameters such as diel growth pattern and total lipid content produced to conclude whether the generated lines can be useful for breeders. Hence, there is a long way of new experiments and modifications to perform to get a good story to propose for publication.
Author Response
Response to Reviewer 2
12/31/2022
First, I would like to thank the reviewer for the comments on the manuscript. For your convenience, all my responses are in blue color text.
Open Review
(x) I would not like to sign my review report
( ) I would like to sign my review report
English language and style
( ) English very difficult to understand/incomprehensible
( ) Extensive editing of English language and style required
( ) Moderate English changes required
(x) English language and style are fine/minor spell check required
( ) I don't feel qualified to judge about the English language and style
|
Yes |
Can be improved |
Must be improved |
Not applicable |
|
|
Does the introduction provide sufficient background and include all relevant references? |
( ) |
(x) |
( ) |
( ) |
|
Are all the cited references relevant to the research? |
( ) |
(x) |
( ) |
( ) |
|
Is the research design appropriate? |
( ) |
( ) |
(x) |
( ) |
|
Are the methods adequately described? |
( ) |
( ) |
(x) |
( ) |
|
Are the results clearly presented? |
( ) |
( ) |
(x) |
( ) |
|
Are the conclusions supported by the results? |
( ) |
( ) |
(x) |
( ) |
Comments and Suggestions for Authors
The present manuscript proposes to generate mutant population by EMS and to identify high oleate mutants by GC analysis. This study is based on the use of a specific accession, that has already a “high oleate content” due to possible mutations in the FAD2 gene.
Response: Thanks for the comment. We revised the clarify for this point. We spent over a year to identify this natural mutant (PI 263470) by screening over 1,300 sesame accessions. From literature and germplasm passport, this accession was never identified and reported as high oleate (54% vs 39.5%). This information alone is important for breeding high oleate sesame. Among all accessions we screened, this accession has the highest oleate content. However, the oleate content is lower than the high oleate peanut or soybean. Thus, we set up EMS mutagenesis to see if we can further enhance the oleate content. This manuscript showed that we further improved oleate content. Although the mechanism for the further enhanced oleate is not known now, it is still useful for breeding. Furthermore, we are working toward elucidate the mechanism in our future work.
From our morphological observation (from seedlings to mature plants) at greenhouse for several generations, this accession showed morphological uniformity. From our DNA sequencing and deduced amino acid data (please see Figure S1), we are very sure that the G425A natural mutation (not due to possible mutations) of FAD2 gene is responsible for the high level of oleic acid (from 39.5% to 54%). Similar high oleate mutations were also identified in several other crop species.
Then, EMS mutagenesis is performed, followed by screening and finally the lipid composition of some mutant plants is analysed. While this work can have a significant interest for plant breeders, most of the conclusions draw by the study are not supported by the data. Indeed, “high oleate content” suggest that there is more oleate in terms of quantity. However, all the GC analysis showed % fatty acid profile, i.e. we do not have any information about the lipid quantity. Thus, differences in the fatty acid profile do not necessarily imply final differences on the fatty acid content.
Response: I understand your concerns on lipid quantity. The percentage of oleate in the total lipids reflect the relative oleate level and represent the true quality of the oil and the relative quantity of oleate in the total oil. Although it is possible to determine the absolute quantity of oleate per seed or gram of seeds on HPLC, but it is highly variable even with an internal standard. Since our focus is on the relative amount (percentage) of oleate in the total fatty acids, we believe our conclusions are well supported by our data.
In fact, lipid analysis for fatty acid is always shown in terms of % and of quantity to disentangle this issue. Hence, the “high oleate content” promised in the study is questionable. The authors must show the quantitative results for all the mutant lines and discuss about it. Besides this, the authors proposed a link between a single point mutation and the activity of FAD2 with only one plant as A/G and one plant as G/G. I am not so convinced by these results and I think the authors should perform a backcross of an A/A line with a G/G line to obtain multiples lines with A/G pattern to confirm that the fatty acid profile corresponds to a mixture of A/A and G/G fatty acid profiles. This will also help the authors to obtain more biological replicates.
Response: Thanks for the comments. We clarified this point in the revised text. We initially identified an accession with high oleate 54.0% that has not been known to have high oleate. We planted the seeds from this accession in the greenhouse and observed plants with uniformed plant morphology. Then we collected leave samples for DNA isolation and seeds for GC analysis from individual plants. The results were presented in Table 1 and Figure 1. All six single plants with genotype A/A showed high oleate (52.8-56.6%) and one plant with G/A showed 47.2% much lower than other six plants but significantly higher than the average (39.2%, PI 224663). We believe this experiment constitutes a co-segregation analysis. Furthermore, the mutation of G425A in FAD2 gene for the high oleate is also supported by observations from several other crop species (peanut and soybean).
Indeed, the statistical significance of the results can also be questioned: there are no biological triplicates for all the GC analysis, and some student’s t-test are performed with only two values per mean…Finally, the table 1 and 2 should be modified as bargraphs to be able to rapidly compare differences of fatty acid profiles.
Response: Thanks for the comment. In our experiment design, we added the check PI 263470 with A/A. For each genotype, we had 8 plants, which is in fact replicates. First, we would like to compare with the check. Second, we had six mutant plants (six replicates). They can be compared with each other. I think we had enough data for statistical analysis. This experiment design was not ideal, but it was sufficient for comparison analysis. For each trait, the mean value was from eight individual plants. Table 1 was for identification of the genotype and Table 2 was for comparison of each trait among different genotypes in detail. For the significance of each trait and each line, the results were displayed in Figure 5 and 6. I think our results were presented clearly.
Regarding the absence of mutation in the FAD2 gene in the newly established mutant lines, I do not see any surprise. As the authors performed random mutations, many mutations in the genome of sesame can lead to a difference of the fatty acid profile without implying FAD2 gene. The authors also argued that some mutations in the promoter region of the FAD2 gene could occur in the mutant line and perhaps explained the phenotype for the fatty acid profile. Thus, the authors suggest that there is a differential expression of FAD2 gene in the mutant plants along seed filling. So can you please show us qPCR experiments for FAD2 gene in these mutants during seed filling ?
Response: This is a good point. The cause for enhancing the oleate level from 56% to 70% known at present. As the reviewer’s comment, we agree that the most likely reason is additional locus. We need more work to identify the genetic mechanism for the enhanced oleate.
Regarding the metadata about the phenotypical characterization of the mutants, I found it very short. I think it is necessary to have access to important parameters such as diel growth pattern and total lipid content produced to conclude whether the generated lines can be useful for breeders. Hence, there is a long way of new experiments and modifications to perform to get a good story to propose for publication.
Response: This study was a continuation of our identification of high oleate mutant in the USDA sesame germplasm. Enhancing the level of oleic acid was our focus. I agree that there is a long way to go for developing a new sesame cultivar with high oleate trait, but these mutant lines are valuable breeding materials to help sesame breeders and geneticists to breed high oleate varieties and conduct genetic studies.
Reviewer 3 Report
Dear authors
Your manuscript reports interesting results. Below you will find a good number of questions to be answered and issues to be corrected. However, the text needs to be thoroughly reformulated and rewritten in a very concise (e.g. non repetitive) form (e.g notice the number of times that the number of M2 plants appear, or the repetitive explanation of very small moments of the plant cultivation procedure ) . Although not existing a formal “Discussion”, the results need to be better discussed. Multiple previous studies on mutagenesis on sesame are not referred, neither discussed.
Some (only some) examples of the issues that need your attention.
Line 18 . .” 635 M2 mutant plants “. Why “mutants”? I suppose that they are only “M2 plants”, I suppose that no mutations were identified at the phenotypic or genotypic level for all the referred to 635 plants.
Line 43 - Significant variation for oil%, protein%. I suppose that these are orthographic errors Please correct!
In some places: (suggestion) “cultivars/trait” to be replaced by “genotypes”.
Lines 69 – 71 . (suggestion) Screening the germplasm collections and EMS- mutagenesis are two efficient approaches, respectively, for identification of natural occurring or induction of new high oleate mutants. and creation of a newly induced high oleate mutant
Line 74 – I suppose that the objective was to identify or to obtain mutants producing “over” a determine value, but not an exact value (80%).
Lines 106 – Please refer to here to the mutation (DNA) that leads to the amino acid change, and erase the reference to this mutation in the next phrase (redundance). Please, rephrase the rest of the paragraph.
Line 106 – 112 – Why this “functional mutation” reduces the “fatty acid desaturase activity” ? There is any comparative biochemical test of this mutant enzyme vs. the most common? If not, you only “assume” or “suppose”.
Line 121- The word “suggest” is correct, since the identified mutation can be genetically linked to another (unknown) mutation (a biochemical test of cloned common (wild) and mutated enzyme could be decisive).
FIGURE 1 – NOTICE THAT HAS NO CAPTION. Better if the results for the “mixed genotypes” are substituted by those of the genotype G/G.
Line 130 – 0% concentration is “no” concentration. Refer to your control Controls, Blanks?) (phosphate buffer or water, or both).
Lines 144- 145 “Only 329 M2 lines produced M3 seeds. The M2 seed setting rate (31.94%) 144 was probably reduced due to the homozygosity after EMS treatment”. Dear authors . WHAT IS THIS? What is the relationship between an EMS treatment and “homozygosis”?. Mutagens are, obviously, mutagenic and, also, toxic and cytotoxic.
Please do not refer to issues as allowing some seeds to be lost during washing, etc. Please rewrite the Material and Methods clear and concise.
In general terms is not clear (for me) if you are using a “pedigree” or “bulk method” from generation to generation (THIS NEED TO BE CLARIFIED!!). Reading the manuscript it seems that the authors have not sow the seeds using a pedigree approach (e.g. sowing the M2 seeds family by family, apart)-
For example (LINE 154) there are not M2 lines but M2 families (seeds produced by one M1 plant).
Lines 160 – 162. “Since identification of high oleate mutants was our focus in this study, only a few M3 mutant lines were planted for further morphological observations in the early generations” Why are authors referring to this? What happens to these M3 families?.
Line 165 – “M3 seeds”. What is this exactly? M3 seeds are seeds produced by M2 plants. Seeds of M3 plants are M4 plants!
Lines- 164 – 165. Here, the number of plants (M2 or M3) which seeds were analyzed need to be referred to.
Line 173 . “Two possible …” This should be the start of a (clearly separated) new paragraph! Notice that these working hypotheses made sense to be transferred to the “Discussion” and in “Conclusions” as further prospects.
Table 2 – Notice that you have included average values. The title is not clear. One solution is to reduce the size of the numbers to lace the extreme values within parenthesis not divide bay a line, and to differentiate very clearly (not with a slash) between the horizontal genotypes and the vertical values.
Line 230 – “negatively correlated (??) change between oleic acid and linoleic acid”-
LINES - 260 – 261 – VERY IMPORTANT “80ËšC/8 hours for day with natural light and 90ËšC/16 hours with natural dark”. PLEASE CONVERT FAHRENHEITS TO CELSIUS. At the referred temperatures everything will be fried.
The text should be more concise and straightforward. Check, for example Lines 265 to 268: “The seeds were harvested from each capsule. These seeds were put into small paper bags. The paper bags were transferred to the seed storage room for drying (21ËšC, 25% RH for two weeks). The dried seeds were used for next generation planting or fatty acid analysis.”
Lines 278 -280 – I would recommend the EMS solution to be transferred to new vial and decontaminated with NaOH, and seeds to be washed (immersed) with tap water (3-4 changes, during 30 min.), which should also be decontaminated and transferred to a waste container.
Please eliminate the last phrase: “Our results demonstrate that screening germplasm accessions and EMS mutagenesis are efficient approaches for sesame improvement”. This is true for all crops, and if this was the main conclusion of your work (which is not the case), it will not deserve to be published in any modern international journal (my reviewer opinion).
Best Regards
Author Response
Response to Reviewer 3
12/31/2022
First, I would like to thank reviewer’s comments and suggestions for enhancing the manuscript quality. For your convenience, all my changes would be in blue color text.
Review Report Form
Open Review
( ) I would not like to sign my review report
(x) I would like to sign my review report
English language and style
( ) English very difficult to understand/incomprehensible
( ) Extensive editing of English language and style required
(x) Moderate English changes required
( ) English language and style are fine/minor spell check required
( ) I don't feel qualified to judge about the English language and style
|
Yes |
Can be improved |
Must be improved |
Not applicable |
|
|
Does the introduction provide sufficient background and include all relevant references? |
( ) |
( ) |
(x) |
( ) |
|
Are all the cited references relevant to the research? |
( ) |
( ) |
(x) |
( ) |
|
Is the research design appropriate? |
(x) |
( ) |
( ) |
( ) |
|
Are the methods adequately described? |
( ) |
( ) |
(x) |
( ) |
|
Are the results clearly presented? |
( ) |
( ) |
(x) |
( ) |
|
Are the conclusions supported by the results? |
( ) |
(x) |
( ) |
( ) |
Comments and Suggestions for Authors
Dear authors
Your manuscript reports interesting results. Below you will find a good number of questions to be answered and issues to be corrected. However, the text needs to be thoroughly reformulated and rewritten in a very concise (e.g. non repetitive) form (e.g notice the number of times that the number of M2 plants appear, or the repetitive explanation of very small moments of the plant cultivation procedure) . Although not existing a formal “Discussion”, the results need to be better discussed. Multiple previous studies on mutagenesis on sesame are not referred, neither discussed.
Some (only some) examples of the issues that need your attention.
Line 18 .” 635 M2 mutant plants “. Why “mutants”? I suppose that they are only “M2 plants”, I suppose that no mutations were identified at the phenotypic or genotypic level for all the referred to 635 plants.
Response: Thank you for your correction. I made changes in the abstract and text.
Line 43 - Significant variation for oil%, protein%. I suppose that these are orthographic errors Please correct!
Response: Based on your suggestion, I changed them to oil content and protein content.
In some places: (suggestion) “cultivars/trait” to be replaced by “genotypes”.
Response: “cultivars/trait” was replaced by “genotypes” throughout the text.
Lines 69 – 71 . (suggestion) Screening the germplasm collections and EMS- mutagenesis are two efficient approaches, respectively, for identification of natural occurring or induction of new high oleate mutants. and creation of a newly induced high oleate mutant
Response: Based on your suggestion, the text was changed.
Line 74 – I suppose that the objective was to identify or to obtain mutants producing “over” a determine value, but not an exact value (80%).
Response: Based on your suggestion, the text was changed.
Lines 106 – Please refer to here to the mutation (DNA) that leads to the amino acid change, and erase the reference to this mutation in the next phrase (redundance). Please, rephrase the rest of the paragraph.
Response: DNA mutation was added. The rest of the paragraph was rephrased.
Line 106 – 112 – Why this “functional mutation” reduces the “fatty acid desaturase activity” ? There is any comparative biochemical test of this mutant enzyme vs. the most common? If not, you only “assume” or “suppose”.
Response: Assumedly was added.
Line 121- The word “suggest” is correct, since the identified mutation can be genetically linked to another (unknown) mutation (a biochemical test of cloned common (wild) and mutated enzyme could be decisive).
Response: Thank you for your comments here.
FIGURE 1 – NOTICE THAT HAS NO CAPTION. Better if the results for the “mixed genotypes” are substituted by those of the genotype G/G.
Response: The caption and figure legend were added for Figure 1.
Line 130 – 0% concentration is “no” concentration. Refer to your control Controls, Blanks?) (phosphate buffer or water, or both).
Response: “0%” was changed with “phosphate buffer only”.
Lines 144- 145 “Only 329 M2 lines produced M3 seeds. The M2 seed setting rate (31.94%) 144 was probably reduced due to the homozygosity after EMS treatment”. Dear authors . WHAT IS THIS? What is the relationship between an EMS treatment and “homozygosis”?. Mutagens are, obviously, mutagenic and, also, toxic and cytotoxic.
Response: Sorry for not expressing my meaning clearly. Here I mainly mean M2 reveals more recessive lethal alleles from selfing. This sentence was rephrased.
Please do not refer to issues as allowing some seeds to be lost during washing, etc. Please rewrite the Material and Methods clear and concise.
Response: “seeds to be lost during washing” was removed. Materials and Methods were rewritten.
In general terms is not clear (for me) if you are using a “pedigree” or “bulk method” from generation to generation (THIS NEED TO BE CLARIFIED!!). Reading the manuscript it seems that the authors have not sow the seeds using a pedigree approach (e.g. sowing the M2 seeds family by family, apart)-
Response: Yes. You are right. From M3 seed GC results, we selected M2 plants. Later, we used single plant selection (like single seed descent, SSD). If the plants produced high oleate seeds, we followed them. If not, we stopped to follow them. This served our purpose well for creation of new high oleate mutants.
For example (LINE 154) there are not M2 lines but M2 families (seeds produced by one M1 plant).
Response: Yes.
Lines 160 – 162. “Since identification of high oleate mutants was our focus in this study, only a few M3 mutant lines were planted for further morphological observations in the early generations” Why are authors referring to this? What happens to these M3 families?.
Response: “a few M3 mutant lines” was changed to “a few mutant families. This paragraph was mainly to show some morphological changes after mutagenesis.
Line 165 – “M3 seeds”. What is this exactly? M3 seeds are seeds produced by M2 plants. Seeds of M3 plants are M4 plants!
Response: Sorry. M2 seeds grew into M2 plants but would produce M3 seeds. M3 seeds grew into M3 plants and would produce M4 seeds. I hope my explanation here is clear. Thanks.
Lines- 164 – 165. Here, the number of plants (M2 or M3) which seeds were analyzed need to be referred to.
Response: M3 seeds were analyzed.
Line 173 . “Two possible …” This should be the start of a (clearly separated) new paragraph! Notice that these working hypotheses made sense to be transferred to the “Discussion” and in “Conclusions” as further prospects.
Response: New paragraph started.
Table 2 – Notice that you have included average values. The title is not clear. One solution is to reduce the size of the numbers to lace the extreme values within parenthesis not divide bay a line, and to differentiate very clearly (not with a slash) between the horizontal genotypes and the vertical values.
Response: Table 2 was reformatted and inserted. It looks much better.
Line 230 – “negatively correlated (??) change between oleic acid and linoleic acid”-
Response: Changed.
LINES - 260 – 261 – VERY IMPORTANT “80ËšC/8 hours for day with natural light and 90ËšC/16 hours with natural dark”. PLEASE CONVERT FAHRENHEITS TO CELSIUS. At the referred temperatures everything will be fried.
Response: Thank you spotted the big mistake. The temperature was converted to Celsius.
The text should be more concise and straightforward. Check, for example Lines 265 to 268: “The seeds were harvested from each capsule. These seeds were put into small paper bags. The paper bags were transferred to the seed storage room for drying (21ËšC, 25% RH for two weeks). The dried seeds were used for next generation planting or fatty acid analysis.”
Response: Reworded.
Lines 278 -280 – I would recommend the EMS solution to be transferred to new vial and decontaminated with NaOH, and seeds to be washed (immersed) with tap water (3-4 changes, during 30 min.), which should also be decontaminated and transferred to a waste container.
Response: Reworded.
Please eliminate the last phrase: “Our results demonstrate that screening germplasm accessions and EMS mutagenesis are efficient approaches for sesame improvement”. This is true for all crops, and if this was the main conclusion of your work (which is not the case), it will not deserve to be published in any modern international journal (my reviewer opinion).
Response: This sentence was removed.
Round 2
Reviewer 2 Report
Dear authors,
I am not satisfied by your answers. I can see that no formal modifications were performed regarding my comments.
The manuscript in the present form still contains questionable results/presentations that require revisions. In addition, many parts of the paper should be written again, as suggested by other reviewers.
For example, the abstract do not follow a classical workflow(Background, methods, results discussion).
There are many strange terms used in this manuscript. For example, we use the term "Control" instead of "check" in plant science. Additionnally, the table showed mean values with "range", but where are the standard errors or standard deviations ?
To whom it may concern, the manuscript is almost exclusively based on the table 1 and 2, and this tables are very difficult to read to evaluate oleate content and other fatty acids. Why not to make bargraphs to help the reader ?
Regarding the quantification of fatty acids, it very easy to do it with a internal standard and GC-FID methodology. I made thousands of quantifications with this old method. So the arguments of the authors are largely irrelevant. And why you mentionned HPLC in the answers ?
Regarding the analysis of the A/A, A/G and G/G, there are only two replicates, thus statistical tests cannot be performed. Hence, please had more A/G lines with a segregation test.
I could continue to state some points to ask for modifications, but i do not have enough time to continue to consider this article. Please read your manuscript again and discuss with other scientists from your institute to have some feedbacks. This will undoubtedly help to strengthen the scientific soundness of the study.
Reviewer 3 Report
Dear authors please check the suggestions concerning the "Abstract" (and spirit behind these sugestions) and, please, apply them to the whole manuscript.
The sesame germplasm accession, PI 263470, which contains 55.2% oil and has a significantly higher level of oleic acid (54.0%) than the average (39.5%), was identified by screening the entire USDA germplasm collection. The seeds from this accession were planted in a greenhouse. Leaf tissues and seeds were harvested from individual plants. DNA sequencing of the coding region of (the) fatty acid desaturase gene (FAD2) confirmed that this accession contained a natural mutation of G425A corresponding to the deduced amino acid substitution of R142H leading ( HOW DO YOU KNOW THAT? EVENTUALLY, YOU “SUPPOSE” THAT!) to the high level of oleic acid, but it was a mixed accession with three genotypes (G/G, G/A, and A/A at the position). The genotype with A/A was selected (WHY SELECTED??? Please pay attention because here you are eliminating the G/G and G/A – explain the selection process) and self-crossed for three generations. The purified seeds were used for EMS-induced mutagenesis to further enhance the level of oleic acid. 635 M2 plants were generated from mutagenesis (PLEASE MERGE AND MODIFY THIS PHRASE WITH THE NEXT ONE. PLEASE NOTICE THAT “M2” SELF-EXPLAINS THE ORIGIN OF THE PLANTS). Some mutant plants had significant morphological changes including leafy, flat stems, and others. M3 seeds were used for fatty acid composition analysis by gas chromatography (GC). Several mutant lines were identified with high oleic acid (70%). Six M3 mutant lines plus the check were advanced to M7 or M8 generations. Their high oleate traits from M7 or M8 seeds harvested from M6 or M7 plants were further confirmed (WHY IS THE TEXT GOING BACK AND FORWARD? YOU HAVE IDENTIFIED SOMETHING IN M6 AND M7 WHICH WAS CONFIRMED IN THE NEXT GENERATIONS! Enough!). The level of oleic acid from one mutant line (M7 915-2) was over 75%. The coding region of FAD2 was sequenced from these six mutants, but no mutation was identified. Additional loci may contribute to the high level of oleic acid. The mutants identified in this study can be used as breeding materials for sesame improvement and as genetic materials for forward genetic studies.
TO HAVE THE OLEIC ACID CONTENT ELEVATED TO 75% IS GREAT RESULT BUT, BY ITSELF, THIS RESULT DESERVED A GOOD PUBLICATION IN A USDA TECHNICAL REPORT, NOT IN AN INTERNATIONAL Q1 JOURNAL (MY PERSONAL OPINION, OF COURSE).
PLEASE NOTICE “Additional loci may contribute to the high level of oleic acid.” THIS IS THE MAIN SCIENTIFIC RESULT OF THIS WORK. HOWEVER, THE AUTHORS DO NOT GIVE IT THE NECESSARY IMPORTANCE.
PLEASE REVISE ALL THE TEXT TAKING INTO CONSIDERATION THE ABOVE CONSIDERATIONS,
All the best for 2023
Author Response
Response:
Dear Reviewer,
First, we would like to thank your comments and suggestions for enhancing the manuscript quality.
- The abstract was rewritten. Please see the new version of abstract.
- G425A mutation of FAD2 gene helped us select A/A genotype. This mutation also happened in other species (such as peanut).
- “Check” was replaced with “Control” throughout the text plus figures and tables. All Tables and Figures were reprepared. Based on reviewer’s suggestions, we also converted two tables to two figures. Now we have nine figures, two supplementary figures, and three supplementary tables. The manuscript looks much better than before.
- I totally agree with all reviewers’ suggestions. There are a lot of research work which we need to do, but for this report we do our best to revise our manuscript.
- I researched the references on sesame, but I did not find good ones which are related to this research. Some references were not conclusive or not well supported by their data. Therefore, I only added one references.
I hope that I have addressed all your concerns and questions. Thank you very much for reconsideration of this manuscript.
Sincerely yours,
Ming Li Wang
